# Solvent-Free Fabrication of Thick Electrodes in Thermoplastic Binders for High Energy Density Lithium-Ion Batteries

**DOI:** 10.3390/nano12193320

**Published:** 2022-09-23

**Authors:** Han-Min Kim, Byeong-Il Yoo, Jin-Woo Yi, Min-Jae Choi, Jung-Keun Yoo

**Affiliations:** 1Composites Research Division, Carbon Composites Department, Korea Institute of Materials Science (KIMS), Changwon 51508, Korea; 2Department of Chemical and Biochemical Engineering, Dongguk University, Seoul 04620, Korea; 3Advanced Materials Engineering Division, University of Science and Technology (UST), Daejeon 34113, Korea

**Keywords:** lithium-ion battery, binder, solvent-free, composite material

## Abstract

The rapid development of electric vehicles has generated a recent demand for high energy density lithium-ion batteries (LIBs). One simple, effective way to enhance energy density of LIBs is to increase the thickness of electrodes. However, the conventional wet process used to fabricate thick electrodes involves the evaporation of large amounts of organic solvents, which causes an inhomogeneous distribution of conductive additives and binders. This weakens the mechanical and electrochemical network between active materials, resulting in poor electrochemical performance and structural degradation. Herein, we introduce a new strategy to produce homogeneous thick electrodes by using a dry, solvent-free process. Instead of using a conventional PVDF (polyvinylidene fluoride) binder, we employed a phenoxy resin as the binder in dry process for the first time. This thermoplastic binder exhibits better ductile properties than PVDF in the way that it generates a uniform network structure that connects the active materials during the hot press process. This enables the production electrochemically stable electrodes without using organic solvents, which record capacity retention rates of 73.5% over 50 cycles at a 40 mg/cm^2^ of thick electrodes. By contrast, thick electrodes produced with a PVDF binder via wet processing only have a capacity retention rate of 21.8% due to rapid structural degradation.

## 1. Introduction

High energy density lithium-ion batteries (LIBs) have recently attracted attention with the recent developments of electric vehicles (EVs). To increase the operating distance of EVs, it is essential to fabricate high-energy-density electrodes [1,2,3,4,5]. Thick electrodes (>6 mAh/cm^2^) decrease the stack in LIBs, which effectively increases the overall energy density of LIBs (>350 Wh/kg) and minimizes process expense [6,7,8,9,10]. However, thick electrodes suffer from longer diffusion lengths for Li-ions and electrons, which often generates poorer electrochemical properties [11,12,13,14].

There have been many recent reports that have looked to solve this problem. Ebner et al. reported that the non-uniform pore structure in thick electrodes hinders the penetration of electrolytes, leading to an increase of Li-ion diffusion length [15,16]. The design of porous electrodes with low tortuosity is invaluable in meeting this challenge. For example, in some studies an external electric field was applied to induce a vertical alignment in the electrodes. The resulting 3D stacking of electrodes was reported to precisely control the structure of each electrode layer [17,18,19,20,21]. All these methods conclude that the electrochemical performance of electrode rapidly decreases with an increase of thickness [22,23,24,25]. However, previous approaches are limited to lab-scale fabrication and are difficult to extend to the large-scale fabrication required to meet the demand of industry. Furthermore, slurry-cast electrodes require solvent evaporation processes, which results in a migration of low-density conductive additives and binders [26,27,28]. This phenomenon particularly occurs with PVDF (polyvinylidene fluoride), a binder commonly used in industry, which causes self-aggregation in electrodes and forms non-uniform pores [29,30]. This leads to an inhomogeneous vertical distribution, which inevitably follows the decrease of cyclability and output of LIBs [31].

Herein, we report on a new method to produce thick electrodes that retain their homogeneous distribution. We introduce a solvent-free dry fabrication, using a hot-press and powder composite, consisting of active materials, binders, and conductive additives, which forms thick electrodes on the current collector. Since this dry fabrication does not use organic solvents in the conventional slurry-cast method, it is important to generate a long-range network of binders during the process to form the powder composite. Instead of using a PVDF binder that exhibits low mechanical ductility, poor adhesion with active material, and high processing temperatures (~187 °C), for our study we used a new phenoxy resin binder that has better thermoplastic properties. This phenoxy resin exhibits electrochemically stable properties in the operating voltage window and high plasticity at a relatively low temperature of 90 °C, which provides strong adhesion with active materials and current collectors. This results in a uniform distribution of material composite (LiNi_0.8_Co_0.1_Mn_0.1_O_2,_ phenoxy resin, carbon nanotube) within the thick electrode, which we confirmed using scanning electron microscopy (SEM) and resistance meter. The resultant thick electrode had a high mass loading (~40 mg/cm^2^) and showed excellent cyclability over 50 cycles with a capacity retention rate of 73.5%, whereas thick electrodes with PVDF exhibit retention rates of 8.3%. In addition, thick electrodes produced by the conventional slurry-cast method show a capacity retention rate of 21.8% after 50 cycles. Thus our strategy successfully builds a new type of thick electrode that exhibits better cyclability and retention than those produced via the conventional slurry-cast method or with conventional PVDF binders.

## 2. Materials and Methods

### 2.1. Materials

LiNi_0.8_Co_0.1_Mn_0.1_O_2_ (BASF Shanshan Co., Ltd., China, KY181) was used as received without further processing (Appendix A, Appendix A). Multi-walled carbon nanotubes (BT1003M) were purchased from LG Chem (Seoul, Korea) and dispersed in NMP by Advanced Nano Products Co., Ltd. (Sejong, Korea) A PVDF (polyvinylidene fluoride) binder was obtained from Kureha Co., Ltd. (Tokyo, Japan) phenoxy resin (thermoplastic polymer) pellets were purchased from Gabriel Chem. (The Woodlands, TX, USA)

### 2.2. Thermal Properties

To manufacture a binder film, each binder solution was mixed with a target solid content, poured into a square mold, and dried at 60 °C for 6 h. After that, the free-standing binder film was dried at 90 °C in a vacuum oven for 12 h. The thermomechanical analysis (TMA) of the PVDF and phenoxy resin film (thickness: 150 µm) was performed using a Q400 (TA Instruments, New Castle, DE, USA), with a tension force of 0.05 N, and at a heating rate of 2 °C min^−^^1^ in extension mode. A differential scanning calorimeter (DSC Q2000, TA Instruments, New Castle, DE, USA) was used. The heating run was performed at a constant rate (10 °C/min) from 30 °C to 350 °C for each sample.

### 2.3. Electrical Properties

The resistances of the high energy density electrode (electrode area of 4 cm^2^) were measured using a Hioki electrode resistance meter (Nagano, Japan, XF-057). A constant current (10 mA) was applied to the electrode, and the resulting potential change between the multiprobe was then measured and calculated. Using a powder resistivity measurement system, powder volume resistivity and density were measured at various loads (kgf). The Al foil’s resistance and thickness for calculating the active layer and interface resistances of electrodes were 2.65 × 10^−^^6^ cm^−^^1^ and 20 μm, respectively. Scanning electron microscopy (SEM) images were observed using a JSM-6700F (Japan, JEOL) under an accelerating voltage of 10 kV. The electrodes were treated with a cross-section polisher using IB-19520CCP (Japan, JEOL) under an accelerating voltage of 4 kV to obtain CP-SEM (cross-section polisher) images. Electrochemical impedance spectroscopic analysis was performed at frequencies of 0.1 MHz to 0.1 Hz with a voltage amplitude of 0.01 V using a ZIVE SP1 electrochemical workstation (WonA Tech, Seoul, Korea).

### 2.4. Fabrications of Electrodes and Electrochemical Characterization

To compare the electrodes produced by the wet process and the dry process, the slurries were prepared by high shear mixing the electrode materials at specified ratios of LiNi_0.8_Co_0.1_Mn_0.1_O_2_ (NCM811), CNT (Carbon Nano Tube) dispersion solution (1.5 wt% vs. NCM811), and each binder solution (3 wt% vs. NCM811), at 3000 rpm for 15 min. The doctor-blade-coated electrode on Al foils were dried in an oven at 85 °C for 4 h and then dried at 120 °C for 12 h in a vacuum oven. To manufacture the composite powder, the slurry was mixed in a planetary mixer at 50 rpm, and dried at 85 °C in a vacuum state to remove the solvent. After that, the composite powder was laminated on Al foil using a hot press at 150 °C, 10 t for 5 min. The mass loading of the wet-process electrode and the dry-process electrode were ~40 mg/cm^2^. To evaluate the electrochemical performance of the electrodes, coin cells (2032 coin), a PP separator (Celgard 2400), a Li counter electrode, and the electrolyte (1 M LiPF_6_ in EC: DMC (ethylene carbonate: dimethyl carbonate), 1:1 by volume) were assembled in a glove box under highly pure Ar. Galvanostatic charge/discharge tests in the voltage range of 3–4.35 V at 25 °C were performed at C rates (C/10, C/5) for NCM811. The charge/discharge curves were measured for the WBCS 3000 battery tester system (WonA Tech, Seoul, Korea). In addition, CV measurements were performed using coin cells in the voltage range of 3–4.5 V at a scan rate of 0.2 mV s^−1^.

## 3. Results and Discussion

Figure 1a describes the conventional wet process used to fabricate battery electrodes. To produce thick electrodes, a large amount of slurry must be loaded on the current collector. However, during the evaporation process of the solvent, conductive additives and binders are likely to rise to the top of the electrode due to their relatively low density. This causes a non-uniform distribution of materials in the electrode, resulting in poor mechanical and electrochemical properties. To solve this problem, dry processing can be used to produce thick electrodes. In this process, a powder form of the active materials is coated with conductive additives and binders, and is hot-pressed using a press machine, as shown in Figure 1b. The dry process renders a uniform distribution of material through the thick electrode, which provides a superior mechanical and electrochemical network and higher adhesion between the electrode and the current collector.

We began by measuring the electrochemical stability of PVDF and phenoxy resin (thermoplastic polymer) binders using cyclic voltammetry (CV). The voltage scan was 2 mV/sec in the range of 3–4.5 V, as shown in Figure 2a. An oxidation current is observed at 4.2 V in phenoxy resin, but a reduction current is not observed, therefore, the oxidation reaction of phenoxy resin is an irreversible electrochemical reaction. It is assumed that materials, such as HF, are formed by reacting with LiPF_6_ contained in the electrolyte and OH of the phenoxy resin. However, this reaction could be negligible as it will not electrochemically affect the electrode. (<2 μA/g) [32]. The CV curves show that there are no obvious peaks, indicating that both PVDF and phenoxy resin binders are electrochemically stable within the operating voltage of NCM811 batteries. Therefore, a new phenoxy resin binder could be used as cathode binder in LIB.

To estimate the electrochemical properties of PVDF and phenoxy resin binders as a LIB, we fabricated a LIB with those binders using conventional wet processing (Appendix A). The LIBs using phenoxy resin and PVDF both exhibit excellent cyclability over 50 cycles, with a retention rate of ~90% at a current rate of 1C in the 3–4.35 V window. In a high current rate of 5C, an LIB with phenoxy resin shows a retention rate of 79.1%, which is slightly higher than that of an LIB with PVDF (76.8%) (Appendix A). This supports the hypothesis that phenoxy resin operates well as a binder in LIBs.

To estimate the feasibility of a hot-press based dry fabrication process, we performed analyses using differential scanning calorimetry (DSC) and thermomechanical analysis (TMA). The PVDF and phenoxy resin films prepared by thermal polymerization were used. The PVDF binder showed ductile characteristic above T_m_ (Melting temperature) of 160 °C, which is in good agreement with previous reports (Figure 2b) [33]. In the TMA (0.05 N; 2 °C/min), we observed a large amount of dimension change at a temperature of 187.6 °C (Figure 2c). This means that PVDF binder requires a process temperature above 187.6 °C, which is not easy to apply in LIBs. In contrast, the phenoxy resin binder exhibited a T_g_ (Glass-transition temperature) of 87.0 °C, which is quite lower than the PVDF. This allows for a dramatic dimension change above 79.7 °C. Considering composite powder is processed at 85 °C, phenoxy resin could show better thermoplastic behavior than PVDF, resulting in a uniform passivation of CNT on the surface of cathode active materials.

We then investigated powder resistivity of the composites consisting of NCM811, CNT, and the binders, to study the impact of binders in the powder composites. Volume resistivity, a resistivity of powder composites against leakage current, is summarized in Figure 3 and Table 1. The addition of CNT in NCM811 decreases volume resistivity from 4.90 Ω∙cm to 4.43 Ω∙cm, owing to electrically conductive CNT. It is worthy to note that the NCM811 with CNT and phenoxy resin exhibits a higher volume resistivity (19.31 Ω∙cm) than that of NCM811 with CNT and PVDF (15.35 Ω∙cm). This suggests that phenoxy resin enables a more uniform surface passivation of NCM811 compared to PVDF.

To compare the electrical properties of the powder composite in real LIB systems, we measured electrode resistivity of the active layer and the interface after 24 h of wetting with electrolyte. Wet processing and dry processing were used to prepare electrodes by using either PVDF or phenoxy resin binders. The overall resistivity of an electrode is determined by both the active material and interface between current collectors. We expected that the choice of a suitable binder would provide better adhesion with the active material as well as the current collector. Figure 4a,b describes the resistivity of the active layer before and after wetting with electrolyte. The electrodes made by wet processing using phenoxy resin (3.36 Ω∙cm) and dry processing using phenoxy resin (3.48 Ω∙cm) exhibited a slight lower resistivity than the electrode prepared by wet processing using PVDF (3.68 Ω∙cm) and dry processing using PVDF 3.94 Ω∙cm). Interestingly, wet phenoxy resin and dry phenoxy resin showed a drastic decrease of resistivity after wetting with electrolyte to 68.9% and 36.7%, respectively, compared with before wetting. In contrast, wet PVDF and dry PVDF showed an increase of resistivity after wetting with electrolyte to 106.9% and 114.0%, respectively.

We also measured cross-sectional SEM images to confirm the uniform morphology of electrodes using phenoxy resin (Appendix A). The wet phenoxy resin, as shown in Appendix A, displayed uniform distribution of NCM811 without aggregation between CNT and phenoxy resin. However, the wet PVDF in Appendix A showed local aggregation between CNT and PVDF (yellow circle), thus it induced a non-uniform distribution of NCM811 particles. This resulted in the improved resistivity in the active layer, as shown in Figure 4a,b.

Interface resistivity between the electrode and the current collector measured before and after wetting with electrolyte is described in Figure 4c,d. The wet phenoxy resin and dry phenoxy resin showed an average interface resistivity of 0.12 Ω∙cm and 0.26 Ω∙cm, respectively. The interface resistivity of wet PVDF (0.16 Ω∙cm) and dry PVDF (0.33 Ω∙cm) was higher than that of electrodes using phenoxy resin when the same fabrication process was applied. In addition, wet PVDF and wet phenoxy resin exhibited a larger increase of interface resistivity after wetting than dry PVDF and dry phenoxy resin. This is attributed to the fact that dry processing allows uniform distribution of conductive additives and binders along the whole electrode, resulting in minimizing of increasing interface resistivity after wetting in the electrolyte. However, the wet PVDF and dry PVDF had higher electrode thickness change than the wet phenoxy resin and dry phenoxy resin (Appendix A). It is assumed that the electrode thickness change increased after the electrolyte wetting, as non-uniform pores were formed by the non-uniform binder distribution in the electrode. This electrode thickness change resulted in an increasing resistance.

Moreover, we measured electrochemical impedance spectroscopy (EIS) of half-cells, as shown in Figure 5 and Table 2, which indicated that using phenoxy resin exhibits lower resistivity against charge transport compared with PVDF. Warburg impedance refers to the synergetic effect of the diffusion of lithium ions on the electrode/electrolyte interfaces at semi-infinite length, which corresponds to the semi-circle at low frequency (Figure 5). Higher diffusion can also be indicated by the steepness of the Warburg tail slope. The diffusion becomes challenging when the slope is close to a 45° angle with the real impedance (Z′) axis. Unlike PVDF, phenoxy resin electrodes do not self-aggregate in a binder, so a uniform pore would be formed in the electrode. Thus, it is assumed that the pathway of lithium ions and electrons is short, resulting in low resistance.

To verify the uniform distribution in electrodes produced via dry processing, we observed cross-sectional SEM images using energy dispersive X-ray spectrometer (EDS) mapping (Figure 6). We found that carbon elements, indicating the distribution of conductive additives, were concentrated near the top surface for wet PVDF and wet phenoxy resin. In contrast, uniform carbon distribution was observed for dry PVDF and dry phenoxy resin. Ni elements were uniformly distributed for all samples. These results support the claim that dry processing enables the formation of homogeneous components in thick electrodes while wet processing does not.

We then fabricated Li-ion half-cells using cathode electrodes to characterize the electrochemical performance (Figure 7). Thick cathode electrodes enable large amounts of loading in half-cells (~40 mg/cm^2^). The dry phenoxy resin showed an excellent retention rate of 73.5% over 50 cycles at a current rate of 0.1C, whereas wet phenoxy resin exhibited 63.0% retention after 50 cycles. This result suggests that solvent-free dry processing enables the stable operation of LIBs, while conventional wet processing leads to unstable charge and discharge profiles. In the case of PVDF binders, wet PVDF and dry PVDF exhibited poor retention rates of 21.8% and 8.3%, respectively, over 50 cycles. Furthermore, LIBs with PVDF showed a decrease in coulombic efficiency (CE) due to the dissolution of metal and the side reactions of electrolyte (Figure 7 and Appendix A). Considering the >97% of CE in wet phenoxy resin and dry phenoxy resin, we claim that PVDF is not an appropriate binder for thick cathode electrodes due to their self-aggregation and weak adhesion. Furthermore, dry PVDF experienced a capacity drop after the 40th cycle due to the deterioration of the Li metal counter electrodes upon cycling. As shown in Figure 6, the inhomogeneous distribution of materials in wet PVDF and dry PVDF would have caused non-uniform CEI layer formation and non-uniform lithium-ion concentration gradients. It is assumed that the electrode structure was degraded due to side reactions in the electrolyte. This side reaction is caused by the dissolution of transition metals, such as manganese [30,34]. Thus, the coulombic efficiency was low due to the recovery of a new CEI layer. Furthermore, it is assumed that the capacity decreases because the active material has already been degraded. Phenoxy resin could be a promising candidate as a thick cathode electrode for high energy density LIBs. The additional cycle performance at a current rate of 0.2C shows a similar trend with the result in 0.1C (Appendix A). However, since thick electrodes in this paper were evaluated in a half-cell, it is difficult to evaluate the long cycle and rate performance due to the limitation of using Li metal as a counter electrode and the exhaustion of the electrolyte. Future studies could more accurately evaluate these results using a full cell and different counter electrode, such as graphite anode [35,36,37].

## 4. Conclusions

Here we introduce a solvent-free dry processing method to fabricate thick cathode electrodes for high energy density LIBs. Compared with conventional wet processing, dry processing enables the homogeneous distribution of conductive additives and binders, which leads to excellent cyclability in electrochemical performance. We discovered that the thermoplastic properties of the binders is more important in this dry fabrication process. We therefore employed a new phenoxy resin binder that showed a much lower T_g_ (87.0 °C) compared with the conventional PVDF binder (187.6 °C). This allows the formation of fine network of the binder through the whole electrode that exhibits a low resistivity against Li-ion and excellent electrochemical properties. As a result, thick cathode electrodes with high-loads of 40 mg/cm^2^ exhibited stable cyclability over 50 cycles with a retention rate of 73.5%. This work provides a better understanding of the electrochemical reaction in thick cathode electrodes and suggests a new binder candidate for high energy density LIBs.

## Figures and Tables

**Figure 1 nanomaterials-12-03320-f001:**
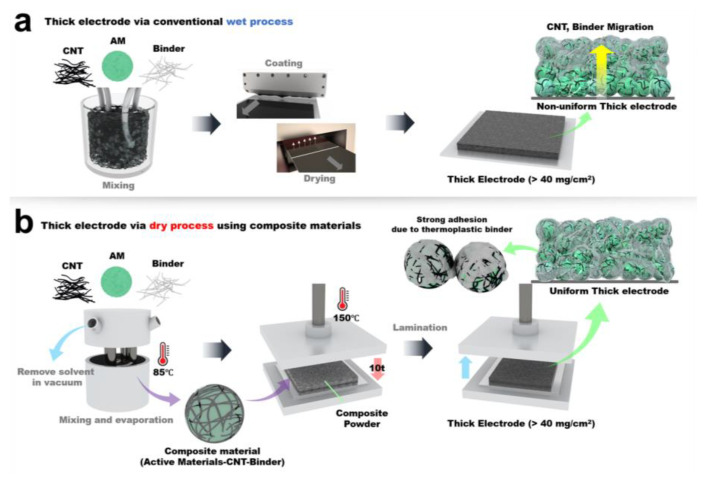
(**a**) Thick electrode produced via conventional wet processing, showing CNT and binder migration during drying; (**b**) Thick electrode produced via dry processing, using composite materials. AM: active materials; CNT: Carbon nano tube.

**Figure 2 nanomaterials-12-03320-f002:**
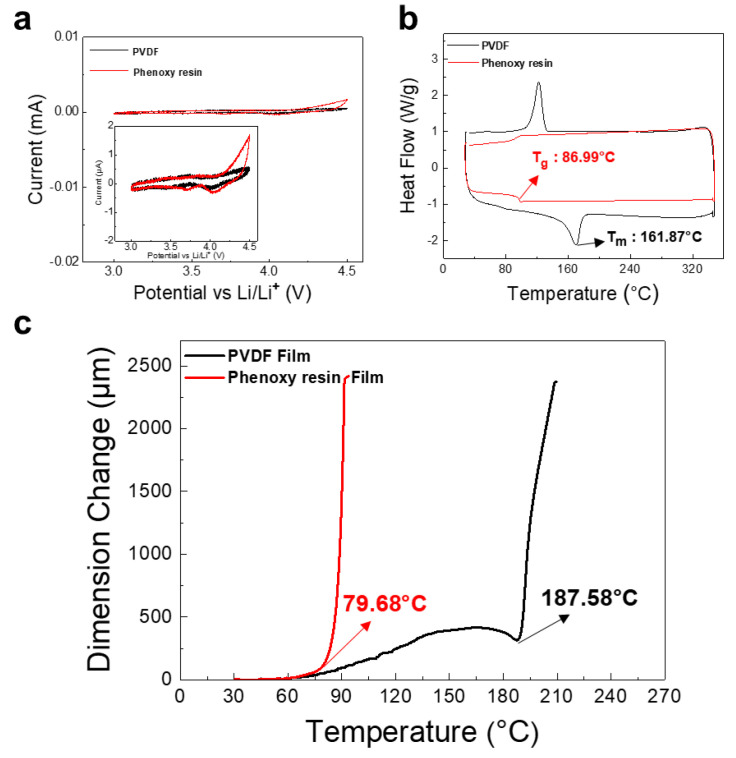
(**a**) CV curves for PVDF and phenoxy resin film measured at a scan speed of 2 mV sec^−1^ in the range of 3–4.5 V (**b**) DSC curves for PVDF and phenoxy resin powder measured at a heating rate 10 °C/min in a temperature range of 30–350 °C (**c**) Thermal mechanical analysis (TMA) of PVDF film and phenoxy resin film in the temperature range of 30–240 °C.

**Figure 3 nanomaterials-12-03320-f003:**
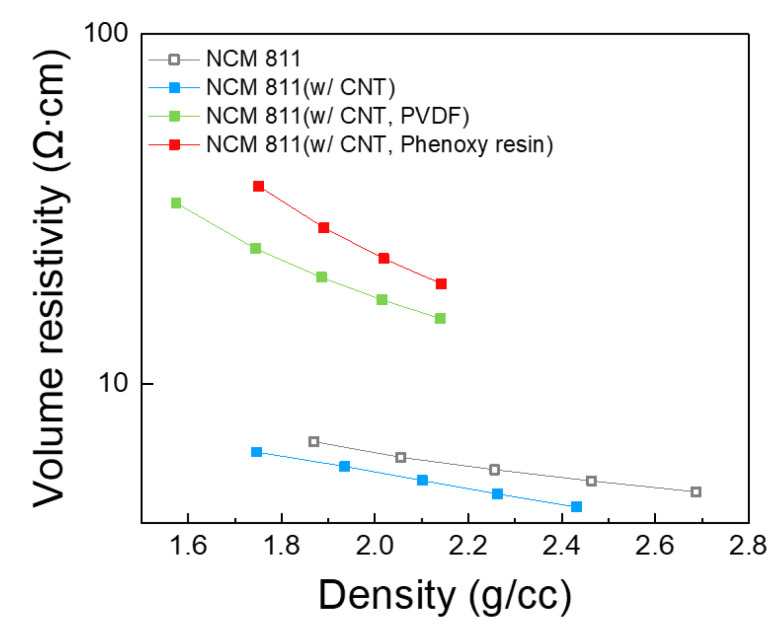
Volume resistivity (Ω.cm) of (NCM811, NCM811 (w/CNT), NCM811 (w/CNT, PVDF), NCM811 (w/CNT, phenoxy resin).

**Figure 4 nanomaterials-12-03320-f004:**
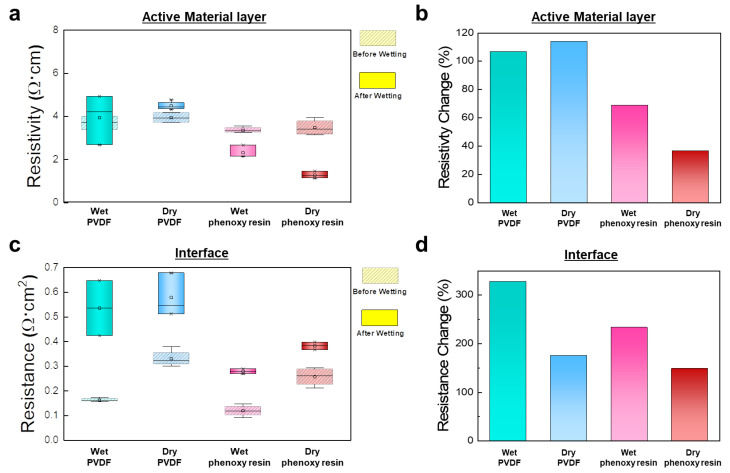
Resistivity of the active material (**a**) and Resistance of the interface (**c**) of the wet PVDF, dry PVDF, wet phenoxy resin, and dry phenoxy resin. Resistivity change of the active material (**b**) and Resistance change of the interface (**d**) of the wet PVDF, dry PVDF, wet phenoxy resin, and dry phenoxy resin.

**Figure 5 nanomaterials-12-03320-f005:**
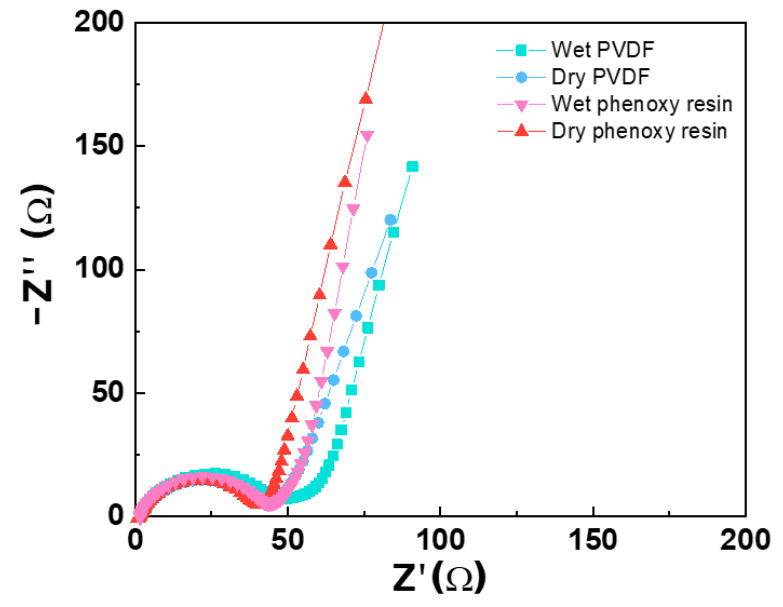
Characterization of electrochemical properties of a half-cell: electrochemical impedance spectroscopic measurement (EIS). Z′: real impedance; -Z″: imaginary impedance.

**Figure 6 nanomaterials-12-03320-f006:**
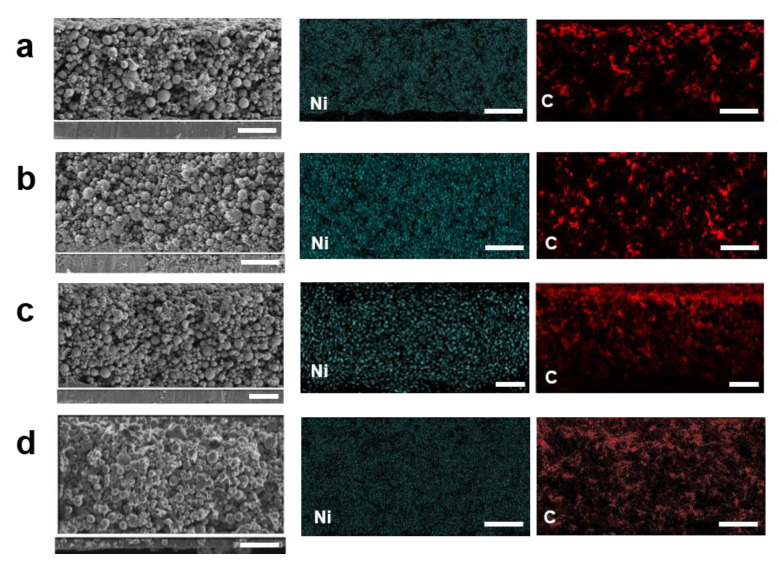
Cross-sectional SEM image and EDS mapping of electrode surface of (**a**) wet PVDF, (**b**) dry PVDF, (**c**) wet phenoxy resin, and (**d**) dry phenoxy resin. Scale bar = 50 μm. Ni and C elemental distribution are indicated by cyan and red, respectively.

**Figure 7 nanomaterials-12-03320-f007:**
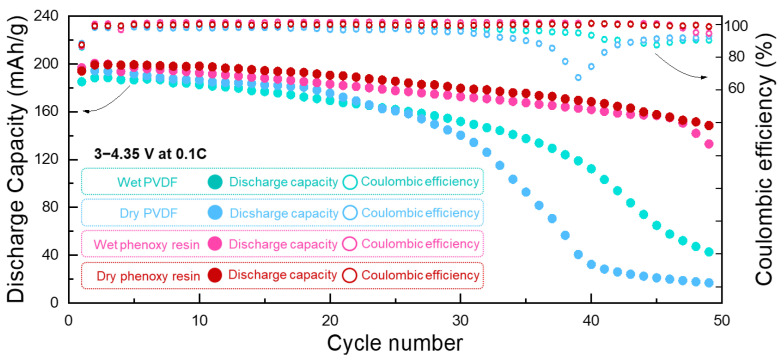
Cycle performances of wet PVDF, dry PVDF, wet phenoxy resin, and dry phenoxy resin at 0.1C.

**Table 1 nanomaterials-12-03320-t001:** Volume resistivity (Ω.cm) of NCM811, NCM811 with CNT, NCM811 with CNT and PVDF, and NCM811 with CNT and phenoxy resin at each density (g/cc).

	NCM811	NCM811 with CNT	NCM811 with CNT and PVDF	NCM811 with CNT and Phenoxy Resin
Volume resistivity (Ω∙cm, @1 g/cc)	4.90	4.43	15.35	19.31
Density (g/cc)	2.68	2.43	2.14	2.14

**Table 2 nanomaterials-12-03320-t002:** Characterization of electrochemical properties of a half-cell: electrochemical impedance spectroscopic measurement (EIS). R_s_: resistance of electrolyte; R_ct_: charge transfer resistance.

EIS [Ω·cm]	Wet PVDF	Dry PVDF	Wet Phenoxy Resin	Dry Phenoxy Resin
R_s_	1.80	1.74	1.84	1.54
R_ct_	48.03	44.06	44.41	40.15

## Data Availability

Not applicable.

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
