# Peer review of "Solvent-Free Fabrication of Thick Electrodes in Thermoplastic Binders for High Energy Density Lithium-Ion Batteries"

_nanomaterials, 2022, doi:10.3390/nano12193320_

Round 1
Reviewer 1 Report
The present article demonstrates the effectiveness of using a phenoxy resin as binder in dry process for improving the performance of LIB with thick positive electrode. the manuscript is well be written and series of analytical studies support the idea of the author. Thus, the manuscript should be published after solving following concerns.
1.
In the abstract, the author said, “In contrast, thick electrode with PVDF binder and wet process shows capacity retention rate of 21.8% due to rapid structural degradation”. However, no experimental data support the structural degradation of electrode. Thus, the additional data should be provided and/or the relevant part of the manuscript should be revised.
2.
In Figure 2a, in positive potential region at around 4.2V~4.5V, the Phenoxy resin sample exhibited higher anodic current than PVDF case. The author should mention about this point.
3.
In Figure 6 and Figure S4, the charge/discharge tests were performed at 0.1C and 0.2C. The author should mention or perform the experiment at higher current density condition, such as 1C, because the battery performance such conditions are practically important.
Author Response
Reviewer #1
Below are my comments:
- In the abstract, the author said, “In contrast, thick electrode with PVDF binder and wet process shows capacity retention rate of 21.8% due to rapid structural degradation”. However, no experimental data support the structural degradation of electrode. Thus, the additional data should be provided and/or the relevant part of the manuscript should be revised.
We have added more detailed explanation to the revised manuscript.
(page 9)
“As shown in Figure. 5, inhomogeneous distribution of materials in Wet PVDF and Dry PVDF would have caused non-uniform CEI layer formation and lithium-ion concentration gradient. It is assumed that the electrode structure was degraded due to side reactions in the electrolyte. This side reaction is caused by the dissolution of transition metals, such as manganese.”
- In Figure 2a, in positive potential region at around 4.2V~4.5V, the Phenoxy resin sample exhibited higher anodic current than PVDF case. The author should mention about this point.
We have added more detailed explanation to the revised manuscript
(page 4)
“An oxidation current is observed at 4.2 V of phenoxy resin, but reduction current is not observed. The oxidation reaction of phenoxy resin is an irreversible electrochemical reaction. It is assumed that materials, such as HF, are formed by reacting with LiPF6 contained in the electrolyte and OH of phenoxy resin. However, this reaction could be negligible because it will not electrochemically affect the electrode. (< 2 μA/g).”
Figure 2. (a) CV curves for PVDF and Phenoxy resin film measured at scan speed of 2 mV sec-1 in the voltage range of 3-4.5 V (b) DSC curves for PVDF and Phenoxy resin powder measured at heating rate: 10 ℃/min in the Temperature range of 30-350 ℃ (c) Thermal mechanical analysis (TMA) of PVDF film and Phenoxy resin film in the temperature range of 30-240 ℃
- In Figure 6 and Figure S4, the charge/discharge tests were performed at 0.1C and 0.2C. The author should mention or perform the experiment at higher current density condition, such as 1C, because the battery performance such conditions are practically important.
We have added more detailed explanation to the revised manuscript.
(page 9)
“However, since thick electrodes in this paper were evaluated with coin half-cell, it is difficult to evaluate the long cycle and rate performance due to the limitation of the Li metal used as a counter electrode and the exhaustion of the electrolyte. In future studies, it can be more accurately evaluated with full cell by using counter electrode such as graphite anode. [35-37]”
[New references]
- Lux, S.; Lucas, I.; Pollak, E.; Passerini, S.; Winter, M.; Kostecki, R. The mechanism of HF formation in LiPF6 based organic carbonate electrolytes. Electrochemistry Communications 2012, 14, 47-50.
- Wang, C.; Xing, L.; Vatamanu, J.; Chen, Z.; Lan, G.; Li, W.; Xu, K. Overlooked electrolyte destabilization by manganese (II) in lithium-ion batteries. Nature communications 2019, 10, 1-9.
- Rowden, B.; Garcia-Araez, N. Estimating lithium-ion battery behavior from half-cell data. Energy Reports 2021, 7, 97-103.
- Raccichini, R.; Amores, M.; Hinds, G. Critical review of the use of reference electrodes in Li-ion batteries: a diagnostic perspective. Batteries 2019, 5, 12.
- Zhang, S.S. Is Li/Graphite Half-Cell Suitable for Evaluating Lithiation Rate Capability of Graphite Electrode? Journal of The Electrochemical Society 2020, 167, 100510.

Reviewer 2 Report
This is a good manuscript.
In the abstract, the authors mention 6 mAh per cm2. For NCM811 and we assume 200 mAh per g, we should get the loading somewhere at 12 mg per cm2 (if this reviewer is not mistaken). In the experiment section, the authors mention 40 mg per cm2. Could you elaborate this a little further?
In the EDX map, the Ni distribution on the cathode made by wet method doesn't seem to be as good as the one made by dry method? Any hint?
The cycle life is technically short to make the battery competitive at industrial level, do you have any longer cycling results and rate capability?
Something went wrong with the battery made with "dry PVDF", the coulombic efficiency went low and up again. Any hint?
What is the average particle size of the NCM811 that you are using? Is there any amorphization on the surface or Li deficiency in the materials? It would be great if the authors can supply some analysis (good SEM, EDX, and XRD) on the NCM811 that was used in the manuscript.
This reviewer has some experience of buying related materials from a particular supplier, where we always see some levels of Li deficiency and not so good crystalinity. However, they are very cost effective.
Lastly, the authors should compare their results with the literature. There have been recent work regarding dry cathode fabrication (i.e. Maxwell Tech bought out by Tesla). Please provide some elaboration.
Author Response
Reviewer #2
Below are my comments:
- In the abstract, the authors mention 6 mAh per cm2. For NCM811 and we assume 200 mAh per g, we should get the loading somewhere at 12 mg per cm2 (if this reviewer is not mistaken). In the experiment section, the authors mention 40 mg per cm2. Could you elaborate this a little further?
We think “>6 mAh/cm2” seems to be misleading. We considered “>6 mAh/cm2 (30 mg/cm2 x 200 mAh)” is the standard for thick electrodes, so it is mentioned. In this paper, the electrode is 40 mg/cm2 * 200 mAh/g (active material theoretical capacity) * 0.952 (active material ratio in the electrode) = 7.6 mAh/cm2, which meets the corresponding criteria above.
- In the EDX map, the Ni distribution on the cathode made by wet method doesn't seem to be as good as the one made by dry method? Any hint?
We checked again Fig 6 and update the high-resolution image in the script.
Figure 6. Cross-sectional SEM image and EDS mapping of electrode surface (a) Wet PVDF (d) Dry PVDF (c) Wet Phenoxy resin (d) Dry Phenoxy resin. Scale bar = 50 μm. Ni and C elemental distribution are indicated by cyan and red, respectively.
- The cycle life is technically short to make the battery competitive at industrial level, do you have any longer cycling results and rate capability?
We have added more detailed explanation to the revised manuscript.
(page 9)
“However, since thick electrodes in this paper were evaluated with coin half-cell, it is difficult to evaluate the long cycle and rate performance due to the limitation of the Li metal used as a counter electrode and the exhaustion of the electrolyte. In future studies, it can be more accurately evaluated with full cell by using counter electrode such as graphite anode. [35-37]”
- Something went wrong with the battery made with "dry PVDF", the coulombic efficiency went low and up again. Any hint?
It shows a similar pattern to Figure 3b, reported in this paper. ( DOI: 10.1002/aenm.202001069)
We assumed Dry PVDF underwent capacity drop after the 40th cycle due to the deterioration of the Li metal counter electrodes upon cycling. Because the non-uniform cnt and binder distribution in the electrode causes the lithium-ion concentration gradient. Thus, the coulombic efficiency is low due to the recovery of a new CEI layer. Also, it is assumed that the capacity decreases due to the active material has already been degraded.
We have added more detailed explanation to the revised manuscript.
(page 9)
“Also, Dry PVDF underwent capacity drop after the 40th cycle due to the deterioration of the Li metal counter electrodes upon cycling. As shown in Figure. 6, inhomogeneous distribution of materials in Wet PVDF and Dry PVDF would have caused non-uniform CEI layer formation and lithium-ion concentration gradient. It is assumed that the electrode structure was degraded due to side reactions in the electrolyte. This side reaction is caused by the dissolution of transition metals, such as manganese. [30,34] Thus, the coulombic efficiency is low due to the recovery of a new CEI layer. Furthermore, it is assumed that the capacity decreases due to the active material has already been degraded.”
- What is the average particle size of the NCM811 that you are using? Is there any amorphization on the surface or Li deficiency in the materials? It would be great if the authors can supply some analysis (good SEM, EDX, and XRD) on the NCM811 that was used in the manuscript.
We added analysis information about NCM811, which is a good comment from reviewers, to the supporting file.
Figure S1. SEM image of NCM811 surface. Scale bar : 1μm
Table S1. NCM811 particle analysis information
NCM 811 |
Particle-size distribution (μm) |
Specific surface area (m2/g) |
Excess Lithium (%) |
||||
Dmin |
D50 |
Dmax |
0.46 |
LiOH |
Li2CO3 |
Total |
|
0.7 |
11.3 |
28.9 |
0.269 |
0.140 |
0.555 |
- This reviewer has some experience of buying related materials from a particular supplier, where we always see some levels of Li deficiency and not so good crystalinity. However, they are very cost effective.
We also think this active material has some levels of Li deficiency and not so good crystallinity. So, if we use better grade of active material to electrodes, we can do more accurately and better study.
- Lastly, the authors should compare their results with the literature. There have been recent work regarding dry cathode fabrication (i.e. Maxwell Tech bought out by Tesla). Please provide some elaboration.
Maxwell Tech fabricates dry electrode using PTFE binders to sheet the electrodes through kneading. Although not included in this paper, it was confirmed that the mechanical properties and electrochemical performance were improved by using composite material applied Phenoxy resin we fabricated to the Maxwell Tech dry cathode fabrication. This data is being prepared for our future paper.
[New references]
- Lux, S.; Lucas, I.; Pollak, E.; Passerini, S.; Winter, M.; Kostecki, R. The mechanism of HF formation in LiPF6 based organic carbonate electrolytes. Electrochemistry Communications 2012, 14, 47-50.
- Wang, C.; Xing, L.; Vatamanu, J.; Chen, Z.; Lan, G.; Li, W.; Xu, K. Overlooked electrolyte destabilization by manganese (II) in lithium-ion batteries. Nature communications 2019, 10, 1-9.
- Rowden, B.; Garcia-Araez, N. Estimating lithium-ion battery behavior from half-cell data. Energy Reports 2021, 7, 97-103.
- Raccichini, R.; Amores, M.; Hinds, G. Critical review of the use of reference electrodes in Li-ion batteries: a diagnostic perspective. Batteries 2019, 5, 12.
- Zhang, S.S. Is Li/Graphite Half-Cell Suitable for Evaluating Lithiation Rate Capability of Graphite Electrode? Journal of The Electrochemical Society 2020, 167, 100510.

Reviewer 3 Report
The manuscript is devoted to fabrication and study of the thick NMC811 electrodes using phenoxy resin as a binder. It could not be published in the present form because of the following issues:
1) The text should be carefully proofread.
2) “We introduce a solvent-free dry fabrication which hot-rolls powder composite” – in Figure 1 a hot press is shown, not a hot rolls. It is necessary to clarify and expand the description of the fabrication method. How was the uniform distribution of the powder over the foil surface achieved?
3) “Instead of PVDF binder exhibiting low mechanical ductility, poor adhesion with active material, and high processing temperature (~187 oC),[32]” – Ref. 32 contains slightly different conclusions. What is the “processing temperature”?
4) “The CV curves show that there are no obvious peaks, indicating that both PVDF and Phenoxy resin binder are electrochemically stable within operating voltage of NCM 811”. In fact, Figure 2a clearly shows the anodic process of polymer oxidation, the authors simply used an incorrect scale along the y-axis. Figure and discussion need to be corrected
5) The results associated with the change in resistance after wetting are completely incomprehensible. "These results imply that electrodes using Phenoxy resin have a uniform distribution through the electrode that the distance between NCM 811 particles is shorter with lower tortuosity" – it is not the explanation. What kind of resistance is measured? Why authors did not use impedance spectroscopy? What is the origin of change in the resistance? This piece of work needs to be redone, or rewritten, or both.
6) What about softening of the composite at relatively low temperature? There are certain processes which start at appr. 40oC shown in Fig. 2c. How can a battery with this type of binder withstand a moderate (40-60oC) heating?
Author Response
Reviewer #3
Below are my comments:
- We introduce a solvent-free dry fabrication which hot-rolls powder composite” – in Figure 1 a hot press is shown, not a hot rolls. It is necessary to clarify and expand the description of the fabrication method. How was the uniform distribution of the powder over the foil surface achieved?
We recognized the typo and corrected the manuscript. "hot-press" is the correct expression, not "hot-roll.” To uniformly cast the powder on the foil, the current collector was attached on a thin metal plate, and the powder was cast on the current collector using a mold, and then the loading was controlled by adjusting the height using vibration.
We have added more detailed explanation to the revised manuscript.
(page 9)
“We introduce a solvent-free dry fabrication which hot-press powder composite, consisting of active materials, binders, and conductive additives, to form thick electrode on current collector.”
- “Instead of PVDF binder exhibiting low mechanical ductility, poor adhesion with active material, and high processing temperature (~187 oC),[32]” – Ref. 32 contains slightly different conclusions. What is the “processing temperature”?
We recognized that the reference was incorrectly added so, deleted it from the script. In Figure. 2b and 2c, we confirmed that PVDF film’s dimension change occurred actively at 187 °C. This means that PVDF is only processed at high temperature. So, PVDF is not good material to apply process of coating a binder and cnt on the surface of active material.
- “The CV curves show that there are no obvious peaks, indicating that both PVDF and Phenoxy resin binder are electrochemically stable within operating voltage of NCM 811”. In fact, Figure 2a clearly shows the anodic process of polymer oxidation, the authors simply used an incorrect scale along the y-axis. Figure and discussion need to be corrected.
We have added more detailed explanation to the revised manuscript
(page 4)
“An oxidation current is observed at 4.2 V of phenoxy resin, but reduction current is not observed. The oxidation reaction of phenoxy resin is an irreversible electrochemical reaction. It is assumed that materials, such as HF, are formed by reacting with LiPF6 contained in the electrolyte and OH of phenoxy resin. However, this reaction could be negligible because it will not electrochemically affect the electrode. (< 2 μA/g).”
Figure 2. (a) CV curves for PVDF and Phenoxy resin film measured at scan speed of 2 mV sec-1 in the voltage range of 3-4.5 V (b) DSC curves for PVDF and Phenoxy resin powder measured at heating rate: 10 ℃/min in the Temperature range of 30-350 ℃ (c) Thermal mechanical analysis (TMA) of PVDF film and Phenoxy resin film in the temperature range of 30-240 ℃
- The results associated with the change in resistance after wetting are completely incomprehensible. "These results imply that electrodes using Phenoxy resin have a uniform distribution through the electrode that the distance between NCM 811 particles is shorter with lower tortuosity" – it is not the explanation. What kind of resistance is measured? Why authors did not use impedance spectroscopy? What is the origin of change in the resistance? This piece of work needs to be redone, or rewritten, or both.
4-1 To compare direct electrical properties of the electrodes, we measured resistivities of the active material and the interface between the electrode layer and the current collector (Figure. 4). The total resistance of the electrode is determined by not only the resistivity derived from the environments neighboring active materials but also that attributed to the interface state between the electrode layer and the current collector.
4-2 We measured electrochemical impedance spectroscopy (EIS) of half-cells, as shown in Figure xx and Table xx, indicates that using Phenoxy resin (Wet Phenoxy resin, Dry Phenoxy resin) exhibits lower resistivity against charge transport compared with using PVDF (Wet PVDF, Dry PVDF). The Warburg impedance is referred to the synergetic effect of the diffusion of lithium ions on the electrode/electrolyte interfaces at semi-infinite length, which corresponds to the semi-circle at low frequency. Higher diffusion can also be indicated by the steepness of the Warburg tail slope. The diffusion becomes challenging when it reaches near to 45° angle with the real impedance (Z′) axis.
Figure 5. Characterization of electrochemical properties of half-cell: Electrochemical impedance spectroscopic measurement (EIS).
Table 2. Characterization of electrochemical properties of half-cell: Electrochemical impedance spectroscopic measurement (EIS).
EIS [Ω·cm] |
Wet PVDF |
Dry PVDF |
Wet Phenoxy resin |
Dry Phenoxy resin |
Rs |
1.80 |
1.74 |
1.84 |
1.54 |
Rct |
48.03 |
44.06 |
44.41 |
40.15 |
4-3
We confirmed that the all electrodes thickness increased after wetting the electrolyte. The Wet PVDF and Dry PVDF had higher electrode thickness change than the Wet Phenoxy resin and Dry Phenoxy resin. It is assumed that the electrode thickness change after the electrolyte wetting was increased as the non-uniform pores formed by the non-uniform binder distribution in the electrode. This electrode thickness change resulted in an increasing resistance.
We have added more detailed explanation to the revised manuscript.
(page 7)
“However, the Wet PVDF and Dry PVDF had higher electrode thickness change than the Wet Phenoxy resin and Dry Phenoxy resin. (Table S2) It is assumed that the electrode thickness change after the electrolyte wetting was increased as the non-uniform pores formed by the non-uniform binder distribution in the electrode. This electrode thickness change resulted in an increasing resistance.”
Table S2. Electrode thickness change before/after wetting the electrolyte.
|
Wet PVDF |
Dry PVDF |
Wet Phenoxy resin |
Dry Phenoxy resin |
|
Thickness (μm) |
Before wetting |
150.48 |
152.47 |
149.75 |
150.63 |
After wetting |
164.02 |
167.72 |
155.74 |
155.15 |
|
Thickness Change (%) |
108 |
110 |
104 |
103 |
- What about softening of the composite at relatively low temperature? There are certain processes which start at appr. 40oC shown in Fig. 2c. How can a battery with this type of binder withstand a moderate (40-60oC) heating?
The softening shown at 40-60oC causes shrinkage due to polymerization curing from insufficient thermal polymerization of the binder film. The peakless curve will be measured if this sample is cooled and measured again. We re-measured the TMA by making a new binder film and attached a more accurate result to the script.
Figure 2. (a) CV curves for PVDF and Phenoxy resin film measured at scan speed of 2 mV sec-1 in the voltage range of 3-4.5 V (b) DSC curves for PVDF and Phenoxy resin powder measured at heating rate: 10 ℃/min in the Temperature range of 30-350 ℃ (c) Thermal mechanical analysis (TMA) of PVDF film and Phenoxy resin film in the temperature range of 30-240 ℃
[New references]
- Lux, S.; Lucas, I.; Pollak, E.; Passerini, S.; Winter, M.; Kostecki, R. The mechanism of HF formation in LiPF6 based organic carbonate electrolytes. Electrochemistry Communications 2012, 14, 47-50.
- Wang, C.; Xing, L.; Vatamanu, J.; Chen, Z.; Lan, G.; Li, W.; Xu, K. Overlooked electrolyte destabilization by manganese (II) in lithium-ion batteries. Nature communications 2019, 10, 1-9.
- Rowden, B.; Garcia-Araez, N. Estimating lithium-ion battery behavior from half-cell data. Energy Reports 2021, 7, 97-103.
- Raccichini, R.; Amores, M.; Hinds, G. Critical review of the use of reference electrodes in Li-ion batteries: a diagnostic perspective. Batteries 2019, 5, 12.
- Zhang, S.S. Is Li/Graphite Half-Cell Suitable for Evaluating Lithiation Rate Capability of Graphite Electrode? Journal of The Electrochemical Society 2020, 167, 100510.

Round 2
Reviewer 3 Report
The authors responded to the reviewer's comments, made corrections to the text, and carried out additional experiments. The article can be accepted for publication